# Modelling and Identification of the Hysteretic Dynamics of Inerters

**Ali Siami \* and Hamid Reza Karimi**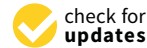

Department of Mechanical Engineering Polytechnic University of Milan, 20156 Milan, Italy;
hamidreza.karimi@polimi.it

**\*** Correspondence: ali.siami@carleton.ca

**Abstract:** This paper deals with an experimental study on the modeling and identification of the hysterical behavior of inerters. Unlike existing methods that can only consider a constant inertance to capture a static model of the device, we develop three different dynamic models for a ball-screw type inerter. To eliminate the effects of the measurement noise, an empirical mode decomposition (EMD) method is proposed. Then, three dynamic models—the Dahl, LuGre and Bouc–Wen model—are used in order to model the friction behavior of the device. Using the least-square optimization method, the parameters of the models are estimated. The results of the tuned models are compared together within different frequencies. The good agreement between predicted and measured data shows that LuGre and Bouc–Wen models can be effective for modelling the hysteretic behavior of friction inside the inerter mechanism. It is also shown that the Bouc–Wen model has better correlation with the experimental results in all test frequencies and amplitudes.

**Keywords:** inerter; friction; hysteresis; empirical mode decomposition; parameter identification

## 1. Introduction

Vibration suppression is a fundamental topic in various systems such as vehicles, wind turbines, buildings and other structures. For this reason, there are intensive studies on various types of devices and suspension systems that can improve the dynamic performance of the isolation systems and suspension systems. In this line, the inerter has received much attention in the literature in recent years. This device was introduced for mechanical mechanisms by Smith [1,2] in 2002. The inerter was first used by McLaren Mercedes for a suspension system in a Formula One race car, then has become the focus of interest in the field. It should be noted that although there are various publications about the application of inerters, they were considered as an ideal device in most of these works.

In this work, a designed and manufactured inerter is tested in order to establish a realistic model for this device. The results of the test which was performed on the device are denoised by using the empirical mode decomposition (EMD) method. Then the parameters of the three different models are estimated using these filtered signals. The hysteretic effect of friction and flexibility of the device is considered in the analytical models which have been presented for the inerter. Based on the authors' knowledge, this type of modeling of an inerter and focusing on the hysteretic behavior of this device is novel and has not been done previously.

There are several papers about the application of this element in the vibration control of different mechanical systems in various types of controlling strategies. For instance, applications of inerters with various configurations to control vibrations in different types of structures have been studied in recent years, (see for instance [3–10]. Shen et al. [11] presented a vehicle suspension system equipped with the inerter and they improved the performance of a dynamic vibration absorber. The parameters of this inerter-spring-damper (ISD) were found by using experimental tests. Then, the identified model of ISD

was used in the numerical model of a suspension system. Palacios-Quiñonero et al. [12] introduced inerter-based multi-actuation system for the seismic protection of adjacent large structures. They used an $H_\infty$ cost function to tune the parameters of tuned mass-inerter damper actuators. The obtained results demonstrate the flexibility and effectiveness of the proposed design methodology.

The dynamics of TMDs with inerters was investigated by Brzeski et al. [13]. The influence of different nonlinearities, such as dry friction and backlash, in the inerter gears on the behavior of TMD was evaluated in this study. The network synthesis method was utilized to determine the structure of ISD suspension system in [14]. The transfer function of a suspension system was obtained by using bilinear matrix inequalities. The performance of the ISD on a quarter car model under random input was verified and it was shown that the root mean square of vehicle body acceleration has been decreased by 18.9%. Chen et al. [15] introduced the inerter to the model of a connected multi-car train system. They showed that the stability and the performance of connected multi-car trains were increased by employing the inerter. In addition, they applied network synthesis methods to realize a mechatronic network and conducted experimental verification. An in-depth experimental study of mechanical devices designed to approximate the dynamics of the ideal inerter was done by Papageorgiou et al. [16]. They focused on experimental identification of physical embodiments of inerters and the study of their departure from ideal behavior. The backlash and its effect of the behavior of the device were considered in mathematical models of two different types of inerters in this work. The inerter nonlinearities, including friction, backlash and the elastic effect, and their impact on vehicle suspension performance were considered in the research of Wang et al. [17]. They built a testing platform to verify the nonlinear behavior of the inerter model. It should be noted that the friction was considered as a constant force in different speeds and frequencies. Sun et al. [18] investigated the effect of ball-screw inerter nonlinearities on the vibration isolation performance of vehicle suspensions systems. They considered friction and the elastic effect of the screw on the modeling part of the device. They obtained the parameters of the model by using recursive least squares algorithm based on the test data. They considered that the friction force was independent from the velocity. The friction force was extracted as a square wave during harmonic tests in low frequency range by the authors. The effect of friction and nonlinear damping force caused by the viscosity of fluid was studied in a type of fluid inerter in [19]. The Coulomb friction model was used to model friction in this device by the authors.

A tuned-mass-damper combined with inerter (TMDI) solution was proposed to improve the performance of an isolation system of the famous statue from Michelangelo Buonarroti named Pietà Rondanini in the vertical direction in [20]. The effectiveness of the proposed method was evaluated numerically. The optimum parameters of the TMDIs were presented by the authors in [21]. It should be noted that the updated multi-degree-of-freedom (MDOF) was presented based on the results of the experimental tests presented in [22]. The effect of a ball-screw type inerter on the dynamic performance of a scaled structure was investigated in [23]. The scaled structure was designed and manufactured according to the results of the experiments which were done on the full scale size of a famous statue. The dynamic performance tests in the presence of the inerter demonstrated the efficiency of this device to improve the dynamic behavior of the suspension system.

In this work, instead of considering an ideal model for inerters, a realistic model including frictional behavior of the device is developed. Based on the results of dynamic performance tests, the parameter of the models are tuned. It should be noted the main focus of this research is to model the hysteretic effect of friction force and to find the most proper and realistic model for this device.

The rest of this paper is organized as follows. In Section 2, a description about the manufactured inerter is presented and then the test setup which is used for experimental tests is discussed. In Section 3, the results of dynamic performance tests are presented. The use of EMD for denoising the measured data is explained in this section and the clean signals are presented. Section 4 is about three different dynamic models for friction. These models are explained in detail in this part. In Section 5, the parameters of the models are identified based on the results of the experiments. Then the results of tuned models

are compared with each other in different working frequencies. The concluding remarks are given in Section 6.

## 2. Modeling the Inerter

In this research, a ball-screw type of inerter is designed. As is shown in Figure 1, in this device, the linear movement of the nut is transformed to rotational movement on the screw and the flywheel by using a ball-screw mechanism. The total inertance of the device is specified by the moment of inertia of the screw and flywheel. Based on the relative acceleration between two terminals of the device, the force on each terminal can be calculated by using the following relation:

$$F_I = b\left(\ddot{x}_1 - \ddot{x}_2\right) \tag{1}$$

where $b$ ($kg$) is the inertance of the device, and $\ddot{x}_1$ and $\ddot{x}_2$ are the acceleration of left and right terminals, respectively. In this kind of inerter, the inertance of the device can be presented as

$$b = \left(\frac{2\pi}{P}\right)^2 J_{total} \tag{2}$$

where $J_{total}$ (kg.m$^2$) is the moment of inertia of the screw and the flywheel along the central axis of the screw and $P$ (m) is the pitch size of the screw. Equation (1) expresses the inerter as an ideal device without considering the effect of friction and flexibility of the mechanical elements. Based on the design of the inerter, the properties of this device are presented in Table 1. The inertance of the device is calculated by Equation (2), considering the assumption of the ideal inerter model. In this study, the inerter is modelled by using the hysteresis effect of friction and considering the inertance of this device. In fact, three different dynamic models are employed to model the dynamic response of the tested inerter.

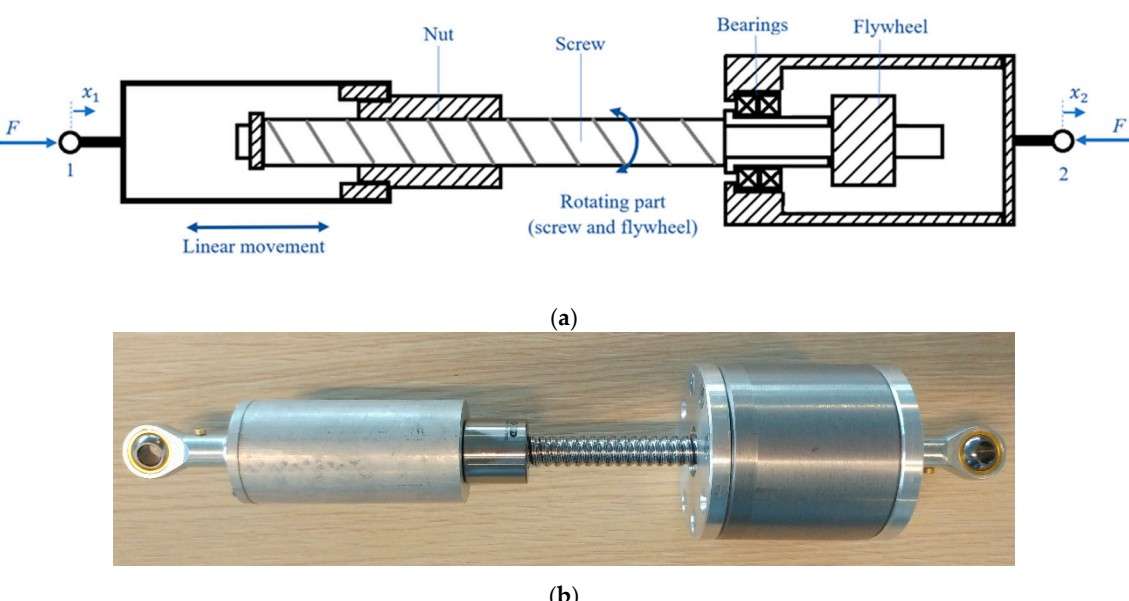

(a)

(b)

**Figure 1.** (**a**) The schematic of the designed inerter and its working concept, (**b**) the manufactured inerter.

**Table 1.** Properties of the manufactured inerter.

| Property | Total Mass (kg) | Moment of Inertia of Rotating Part (kg · m$^2$) | Pitch Size (mm) | Inertance (kg) |
|---|---|---|---|---|
| Value | 2.1 | $6.43 \times 10^{-5}$ | 4 | 158.65 |

In Figure 2, the test setup for the dynamic performance test of the inerter is shown. The inerter is mounted between a load cell and a moving arm which can produce harmonic movement on the lower terminal. This moving part is excited by a pneumatic jack. The load cell measures the force produced by the device on the upper terminal. The harmonic tests were performed in different frequencies between 1 and 4 Hz. In Figure 3, a displacement applied by moving part on the lower terminal of device (see Figure 2) and the measured force on the upper terminal at 1 Hz is plotted.

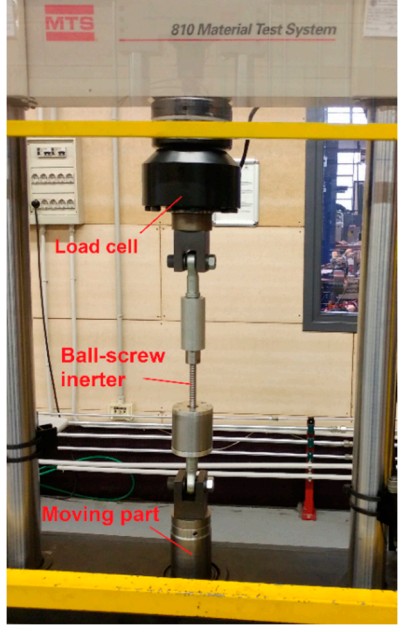

**Figure 2.** Test setup for dynamic performance test of the inerter.

As is clear from the measured force shown in Figure 3, the measured force is very noisy, and the signal needs to be filtered in order to use it for the parameter identification part. Therefore, in the next section, the empirical mode decomposition method will be utilized to denoise the measured forces at different frequencies. It should be noted that the level of displacement imposed on the device varies from 10 mm at 1 Hz to 4 mm at 4 Hz.

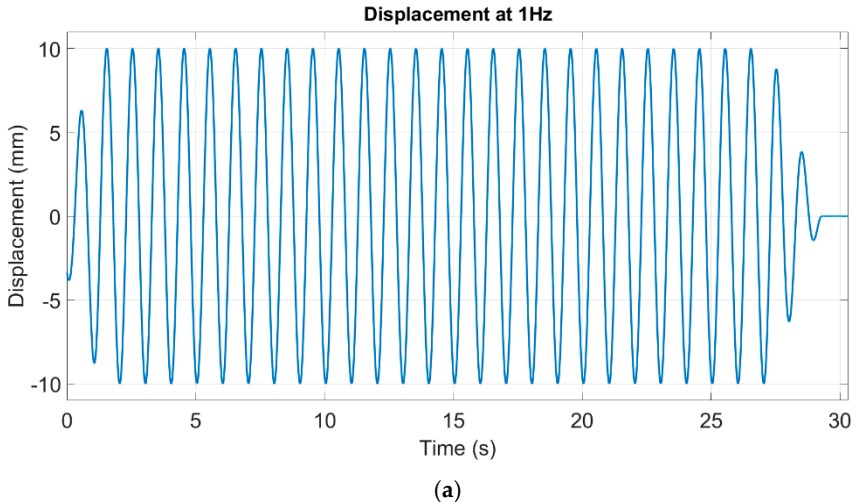

(**a**)

**Figure 3.** *Cont.*

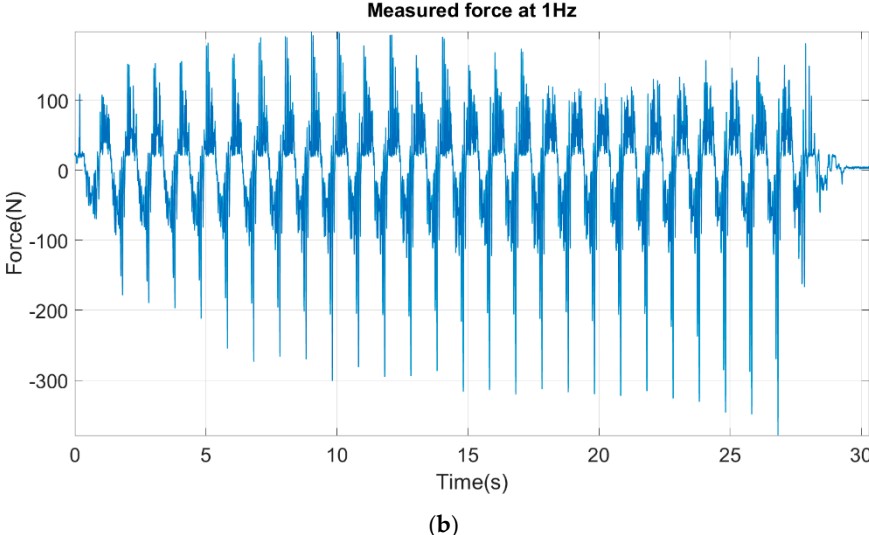

**Figure 3.** Measurement results at 1 Hz; (**a**) displacement imposed to the lower terminal; (**b**) force measured on the upper terminal.

### 3. Denoising Measured Signal by EMD

Empirical mode decomposition (EMD) is a powerful method to remove noise from measured signals in nonlinear and nonstationary [24–26]. In this method, a signal can be decomposed to mono-component signals or an oscillatory mode with one instantaneous frequency, which is called intrinsic mode functions (IMFs). Each IMF needs to satisfy two conditions:

(a)   The number of extrema and the number of zero crossings are either equal to each other or differ at most by one.

(b)   At any point in the time series, the mean value of the upper envelope and the lower envelope is equal to zero.

The process of extracting IMFs by using EMD is called a sifting algorithm. For a signal $x(t)$, the EMD algorithm consists of the following steps:

(1)   Identify the local maxima and minima of the observed signal $x(t)$.

(2)   Interpolate the local extrema using cubic spline to derive the upper and lower envelopes.

(3)   Calculate the median of the envelope $m(t)$ by averaging the upper and lower envelopes.

(4)   Subtract the local mean $m(t)$ from the original signal: $h(t) = x(t) - m(t)$.

(5)   Replace the signal $x(t)$ with $h(t)$ and repeat Steps 1 to 4 until the obtained signal satisfies the two IMF conditions (a) and (b) mentioned earlier. Then, $h(t)$ is an IMF noted as $c(t)$.

(6)   Compute the residue: $r(t) = x(t) - c(t)$.

(7)   Repeat steps 1 to 6 using $r(t)$ for $x(t)$ in order to generate the next IMF and residue.

Then, the original signal can be reconstructed by superposing the obtained IMFs:

$$x(t) = \sum_{n=1}^{N} c_i(t) + r_N(t) \tag{3}$$

where $N$ is the number of IMFs and $r_N(t)$ is the residual.

Based on the mentioned steps for extracting IMFs from the noisy signal, the measured force signals in various tests are decomposed to different IMFs. The first IMFs in each test signal are related to high frequency noise signals. The clean measured force can be found in higher IMFs. For example, for the test performed at 1 Hz, the eighth IMF is the measured force without high frequency noise.

This IMF and the noisy signal are presented in Figure 4. It should be noted that the frequency of this harmonic signal is 1 Hz, which is equal to the frequency of the excitation applied to the test device. The procedure of finding the denoised force signal was applied to the other test signals obtained in different frequencies. The results for 2, 3 and 4 Hz are plotted in Figures 5–7, respectively. These clean signals will be utilized to identify the parameters of different mathematical models for the inerter.

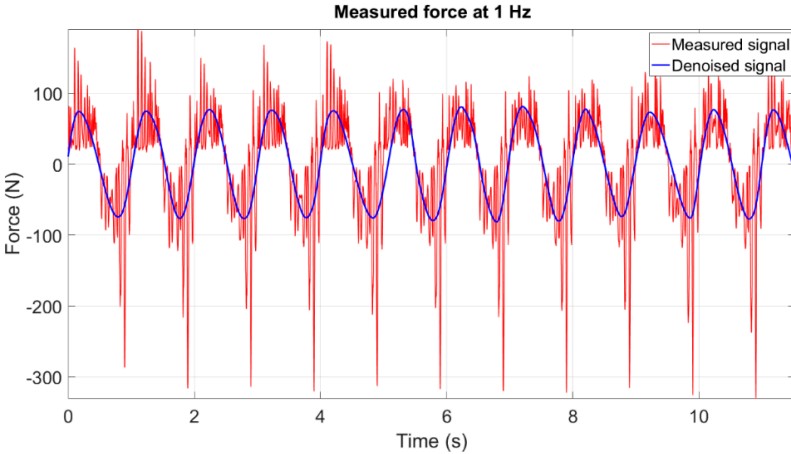

**Figure 4.** Measured force and denoised signal obtained from empirical mode decomposition (EMD) at 1 Hz.

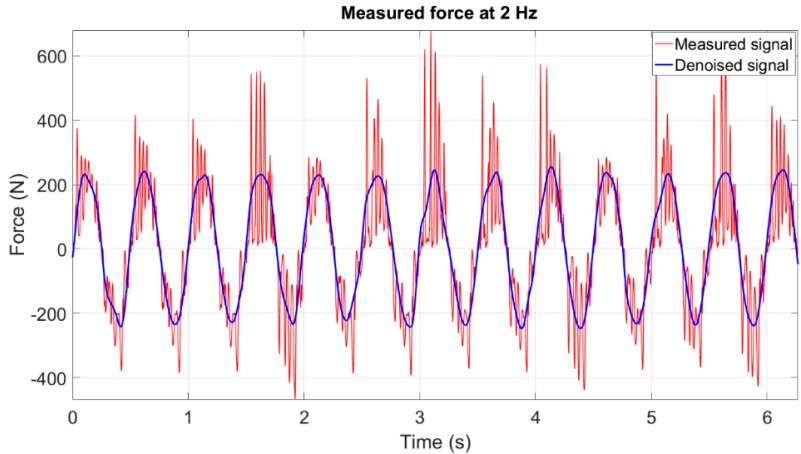

**Figure 5.** Measured force and denoised signal obtained from EMD at 2 Hz.

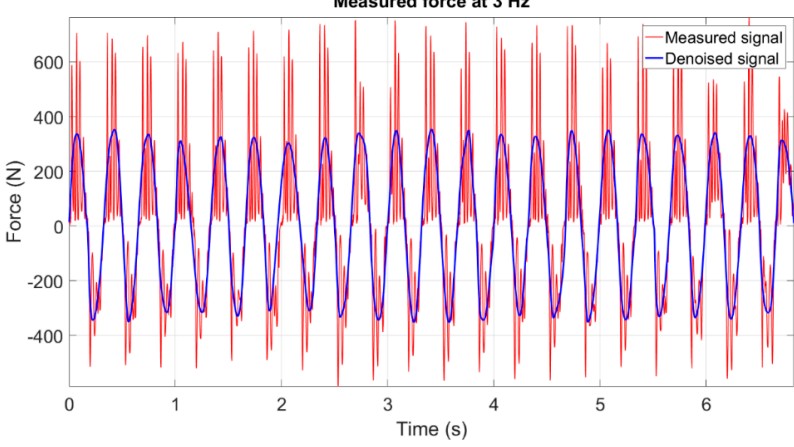

**Figure 6.** Measured force and denoised signal obtained from EMD at 3 Hz.

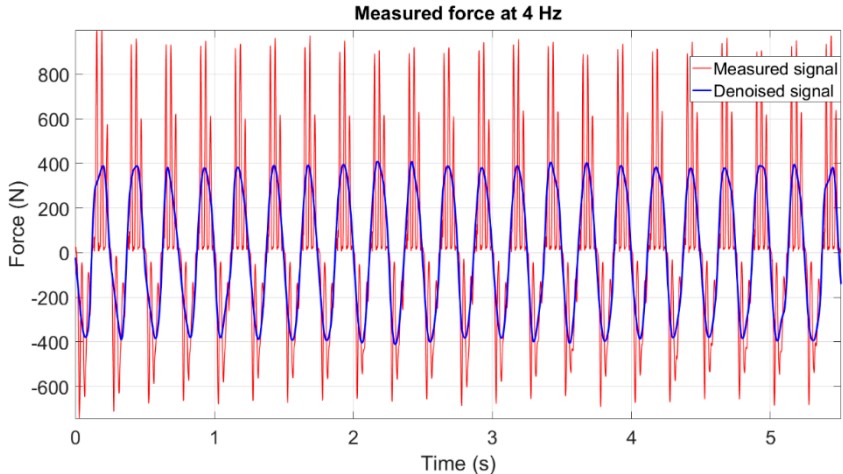

**Figure 7.** Measured force and denoised signal obtained from EMD at 4 Hz.

## 4. Various Models of Friction for Inerter

In this section, different mathematical models are employed for the manufactured and tested inerter. Three models which are used for modeling the hysteretic effect of friction, including Dahl model, LuGre model and Bouc–Wen model, are introduced to the ideal model of the inerter. All models are dynamic models and can cover different features of the frictional behavior of the device. In Figure 8, the ideal model of the inerter and the modified model including the inertance, hysteresis effect of friction, viscous damping and stiffness of the device are presented. The modified model shown in this figure corresponds to the Bouc–Wen model. The details of the three models are presented in the following part.

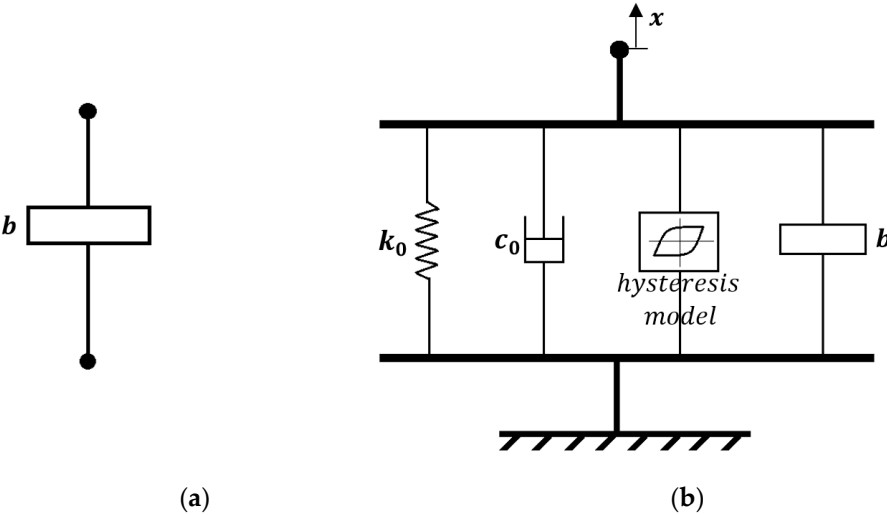

          (a)                    (b)

**Figure 8.** (**a**) Ideal model of inerter, (**b**) modified model of inerter including hysteretic behavior.

The first model that can be categorized as a dynamic friction model was proposed by Dahl [27]. Dynamic friction models assist in obtaining a better approximation of the hysteretic dynamical behavior of the friction force. In Dahl's model, the friction force is undefined when it is less than static friction. In fact, this model developed to describe friction at the pre-sliding stage [28]. The logic behind this model is to utilize the stress–strain curve known from classical solid mechanics, which can be expressed as:

$$\frac{dF_f}{dx} = k_s\left(1 - \frac{F_f}{F_c}sign(v)\right)^{\alpha} \tag{4}$$

where $k_s$ is the stiffness coefficient, and $\alpha$ determines the shape of the hysteresis loop. Introducing $\alpha = 1$ and $F_f = k_s z$, the model can be written as:

$$\frac{dz}{dt} = v - \frac{k_s |v|}{F_c} z \tag{5}$$

$$F_f = k_s z \tag{6}$$

where $z$ is the internal state variable. By introducing this friction term to the equation of an ideal inerter and considering the viscous damping term, the model of the inerter can be modified as:

$$F_I = ba + k_s z + c_s v \tag{7}$$

where $a$ and $v$ are the acceleration and velocity between two terminals of the inerter, respectively, and $c_s$ is the viscous damping coefficient. The inertance of the system represents by $b$. Equation (7) is the Dahl model for the inerter in this paper based on the formulation which has been used to model frictional effect in this device.

The Dahl model does not consider the Stribeck effect and the static friction. Therefore, the LuGre model presented in [28–30] is used here to provide more comprehensive model for frictional effect of the inerter. This model is related to the bristle interpretation of friction [31]. Friction is modeled as the average deflection force of elastic springs. The model is presented as follows:

$$\frac{dz}{dt} = v - \frac{|v|}{g(v)} z \tag{8}$$

$$F_f = \sigma_0 z + \sigma_1 \dot{z} + \sigma_2 v \tag{9}$$

where the internal state variable $z$ denotes the average bristle deflection. The coefficients $\sigma_0$ and $\sigma_1$ are the bristle stiffness and damping, respectively, and $\sigma_2$ is the viscous friction coefficient. It should be noted that variable $v$ is the relative velocity between the contacting surfaces. The function $g(v)$ captures Coulomb friction and the Stribeck effect according to the following formulation:

$$g(v) = F_c + (F_s - F_c) e^{-\left(\frac{|v|}{v_s}\right)^\delta} \tag{10}$$

where $F_s$ and $F_c$ are static friction force and Coulomb fiction force, respectively. The parameter $v_s$ is the Stribeck velocity and it determines how quickly $g(v)$ approaches $F_c$. Using this model for friction, the LuGre model for the inerter can be represented as:

$$F_I = ba + \sigma_0 z + \sigma_1 \dot{z} + \sigma_2 v \tag{11}$$

where $a$ is the relative acceleration between two terminals of the inerter and $b$ is the inertance of the device. The state variable can be calculated by using Equations (8) and (10).

The other dynamic model that it is considered here for modelling friction in the inerter is the Bouc–Wen model. This model is normally used to describe hysterical behavior of materials and elements. The Bouc–Wen model was proposed initially by Bouc [32] in 1967 and subsequently generalized by Wen [33]. This model is one of the most powerful models in order to capture various phenomena related to the hysteretic behavior of friction. The general model's equations are depicted as:

$$\frac{dz}{dt} = \alpha v - \beta |v| z |z|^{n-1} - \gamma v |z| \tag{12}$$

$$F_f = c_0 v + A k_0 x + (1 - A) k_0 z \tag{13}$$

where $\alpha$, $\beta$ and $\gamma$ are parameters that control the hysteresis shape. Coefficient $A$ is a constant that controls the magnitude of friction force. The exponential parameters $n$ assigns the smoothness of transition from the elastic to plastic region. The parameters $c_0$ and $k_0$ are the damping and stiffness coefficient, respectively, and $z$ is the hysteresis displacement of the model. Based on the Bouc–Wen hysteresis model, the model of inerter with inertance equal to $b$ can be presented as:

$$F_I = ba + c_0 v + A k_0 x + (1 - A)k_0 z \tag{14}$$

where the state variable $z$ can be found by Equation (12).

Here we presented three different models which include friction force and its hysteretic behavior. The parameters of these models will be identified in the next section according to denoised test data. Then the results of various model will be compared with each other. It should be noted that these dynamic models were solved numerically in MATLAB in order to obtain the state variable $z$ and then to calculate the force of inerter.

## 5. Parameter Identification and Results

In order to find the parameters of different models, an objective function is defined as follows:

$$P(x) = \left[ \frac{1}{N} \sum_{i=1}^{N} \left( F_{i,exp}(x) - F_{i,mod}(x) \right)^2 \right]^{1/2} \tag{15}$$

where $N$ is the number of data points, $F_{i,exp}$ and $F_{i,mod}$ are the measured force and the force calculated using a specific model, respectively, and $x$ is the variable vector that includes different parameters depending on the model type (Dahl, LuGre or Bouc–Wen model). In the next step, the parameter of the specific model can be found by minimizing the cost function with respect to the vector of variables. Therefore, the optimization problem can be presented as:

$$\underset{x}{\text{minimize}} P(x) \tag{16}$$

where the vector of variables, $x$, is dependent to the type of model of the inerter. For solving the optimization problem, the sequential quadratic programming (SQP) method [34], which is one of the most powerful techniques for the constrained nonlinear optimization problems, is used here. In order to apply this method to the cost function, a MATLAB built-in function was used here. The objective function is the introduced cost function. The vector of variables varies depending on the model which is used for the inerter. The variables for each model, their initial values, lower and upper limits are presented in Table 2.

**Table 2.** The vector of parameters, initial values, lower and upper limits of variables for the different models.

|  | Dahl Model | LuGre Model | Bouc-Wen Model |
|---|---|---|---|
| The vector of variables ($x$) | $[k_s, F_c, c_s, b]$ | $[\sigma_0, \sigma_1, \sigma_2, F_s, F_c, v_s, \delta, b]$ | $[\alpha, \beta, \gamma, n, A, c_0, k_0, b]$ |
| Initial values of the parameters ($x_{int}$) | $[10^5, 1000, 500, 50]$ | $[200, 100, 100, 10, 10, 10^{-3}, 1, 50]$ | $[1, 10, 10, 1, 50, 50, 0.5, 50]$ |
| Lower band if the parameters ($x_{lower}$) | $[0, 0, 0, 10]$ | $[0, 0, 0, 1, 1, 10^{-3}, 10^{-1}, 10]$ | $[1, 1, 1, -5, 1, 1, 1, 250]$ |
| Upper band if the parameters ($x_{upper}$) | $[10^9, 2000, 2000, 250]$ | $[10^5, 2000, 2000, 200, 180, 0.1, 2, 250]$ | $[50, 50, 50, 2, 10^3, 10^3, 10, 250]$ |

Based on the values presented in this table, the results of the SQP method for various models are presented in Table 3. According to this table, the inertance obtained for the device from the LuGre model is 151.5 kg and from the Bouc–Wen model is 154.36 kg. Both values are close to the inertance of ideal model which is 158.6 kg. The inertance obtained from the Dahl model is 174 kg, which it is far from the ideal value.

**Table 3.** The results of optimization problem solution for the various models.

| | Type of the Model | Values |
|---|---|---|
| 1 | Dahl model | $\left[4.29 \times 10^4, 62, 10.2, \ 174\right]$ |
| 2 | LuGre Model | $\left[149.21, \ 443.35, \ 204.38, 1.43, \ 1.0, \ 4.19 \times 10^{-2}, \ 1.98, \ 151.51\right]$ |
| 3 | Bouc-Wen model | $[1.23, \ 1.61, \ 1.35, \ 1.0, \ 341.81, \ 132.00, \ 1.01, \ 154.36]$ |

It should be noted that the results in Table 3 were obtained for the tests which were performed at 4 Hz. The results of three different models identified based on the test data at 4 Hz are plotted in Figure 9. Good correlation between the force of the device obtained from the models and the measured one is clear from this figure. In the next step, these tuned models are used to predict the force of the inerter at 1 Hz. Comparing the results with the test data in Figure 10 shows that only the results of the Dahl model are not fitted to the experimental data. Therefore, this illustrates that using this model is not a proper choice to model friction in the inerter. In should be noticed that at higher frequencies, i.e., at 2 and 3 Hz, these three models have more similar behavior to each other. The force versus time plots at these frequencies are presented in Figures 11 and 12. Based on the force–time plots for the models at different frequencies, it is clear that the LuGre model and Bouc–Wen model provide force responses which are very similar to the measured forces at different frequencies.

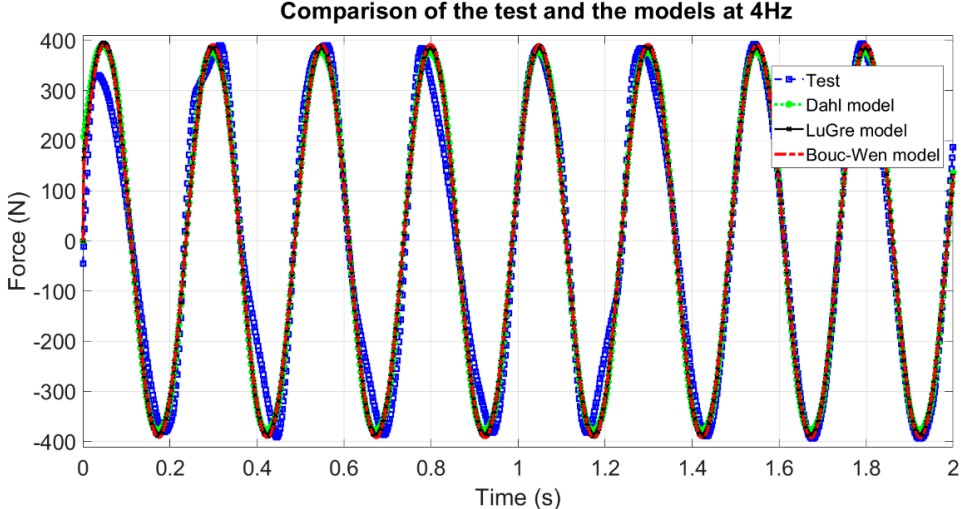

**Figure 9.** Forces obtained from test and various models of the inerter at 4 Hz.

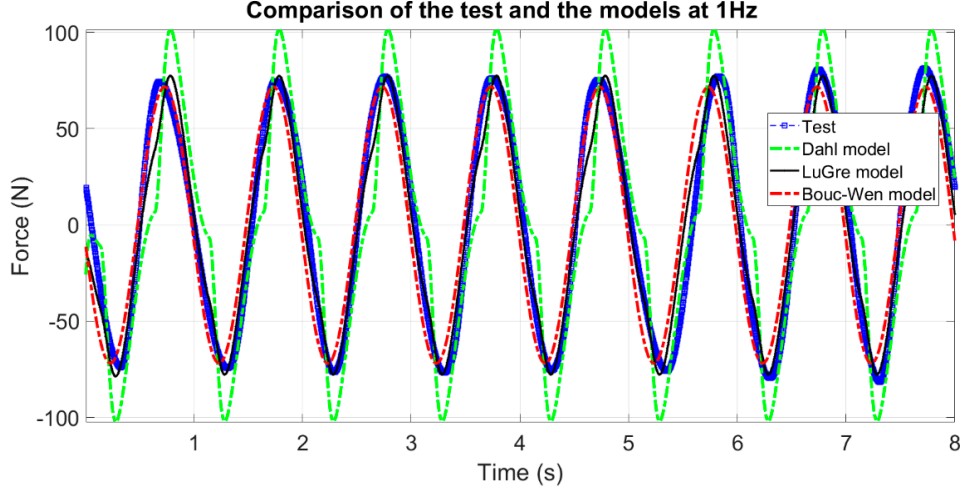

**Figure 10.** Forces obtained from test and various models of the inerter at 1 Hz.

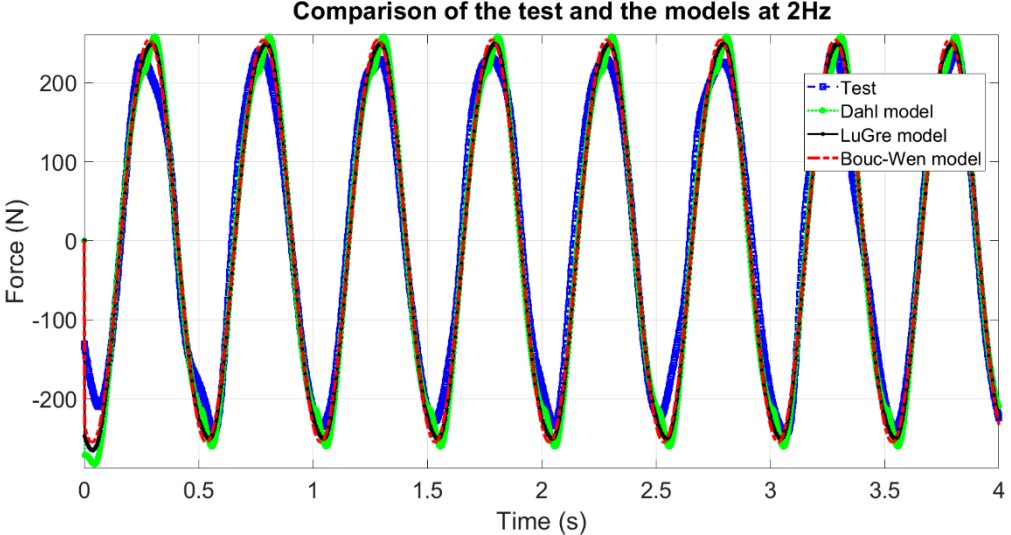

**Figure 11.** Forces obtained from test and various models of the inerter at 2 Hz.

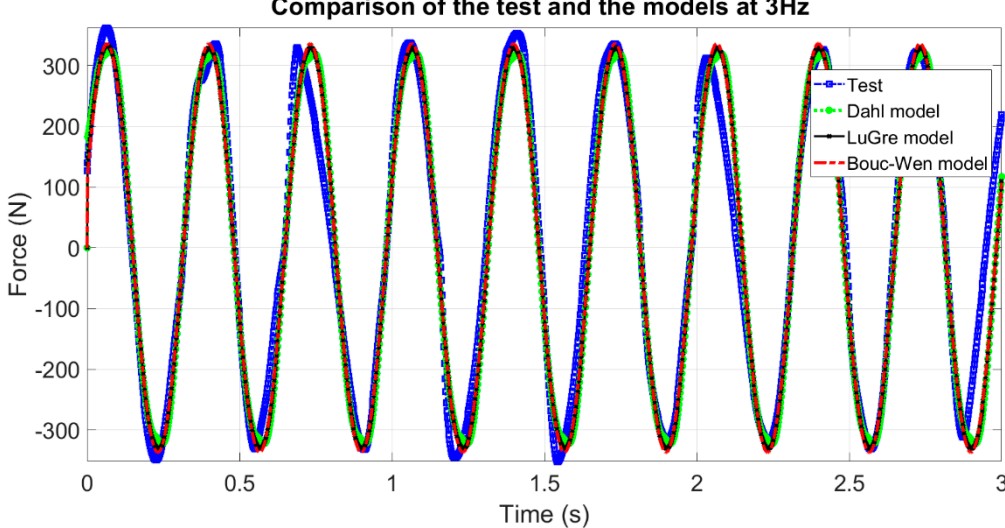

**Figure 12.** Forces obtained from test and various models of the inerter at 3 Hz.

Furthermore, force versus velocity plots at 1 and 4 Hz for various models and for the tests are plotted in Figures 13 and 14, respectively. According to the results presented in Figure 13, it is clear that the Dahl model cannot predict the hysteretic behavior of the inerter in a lower frequency range. In addition, this figure illustrates that the Bouc–Wen model has a better correlation with the test data at 1 Hz in comparison with the two other models. The hysteric loops at 4 Hz in Figure 14 illustrate that at higher accelerations, when the effect of inertance becomes dominant, all three models have good correlation with the test data and they can reproduce the hysteretic loops of the device similar to the experimental tests.

In order to clarify the performance of various models at different frequencies and velocity ranges, the hysteretic loops are presented in Figure 15 for the three models. These plots illustrate that the Dahl model and LuGre model cannot predict the hysteretic behavior of inerter at 1 Hz in comparison with the Bouc–Wen model.

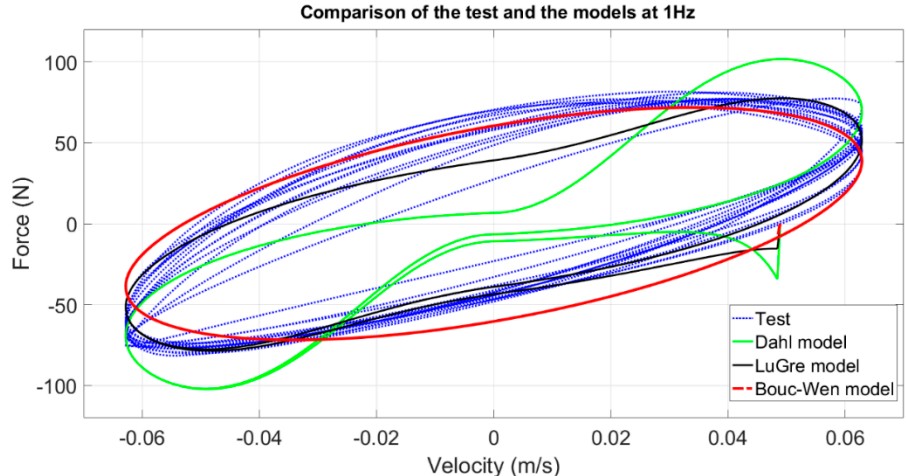

**Figure 13.** Forces vs velocity plot: obtained from test and various models of the inerter at 1 Hz.

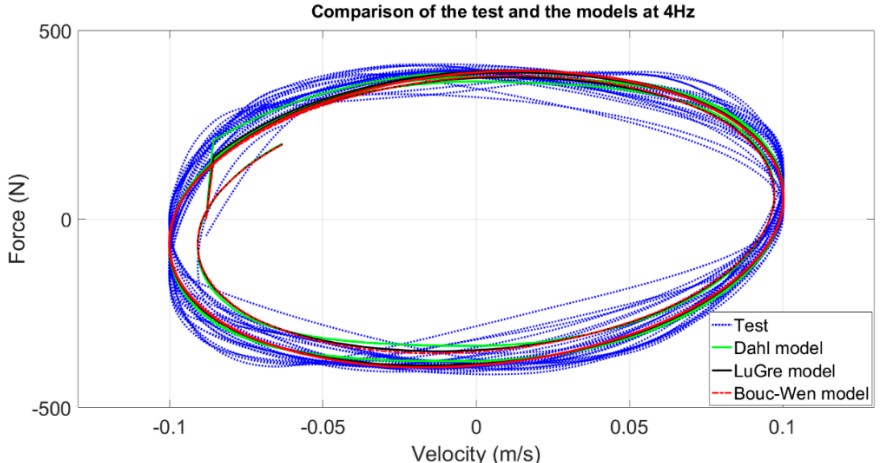

**Figure 14.** Forces vs velocity plot: obtained from test and various models of the inerter at 4 Hz.

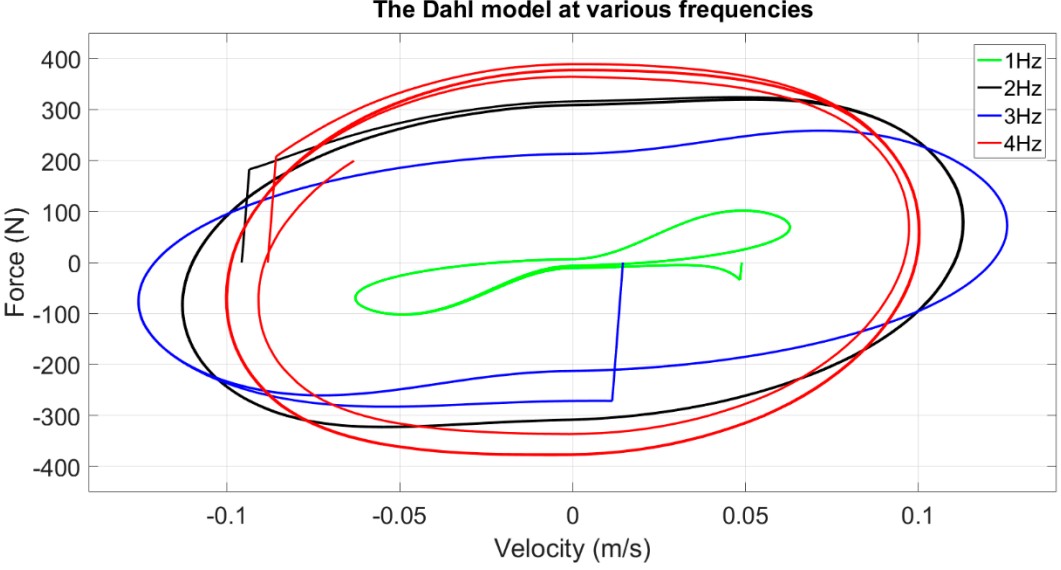

**Figure 15.** *Cont.*

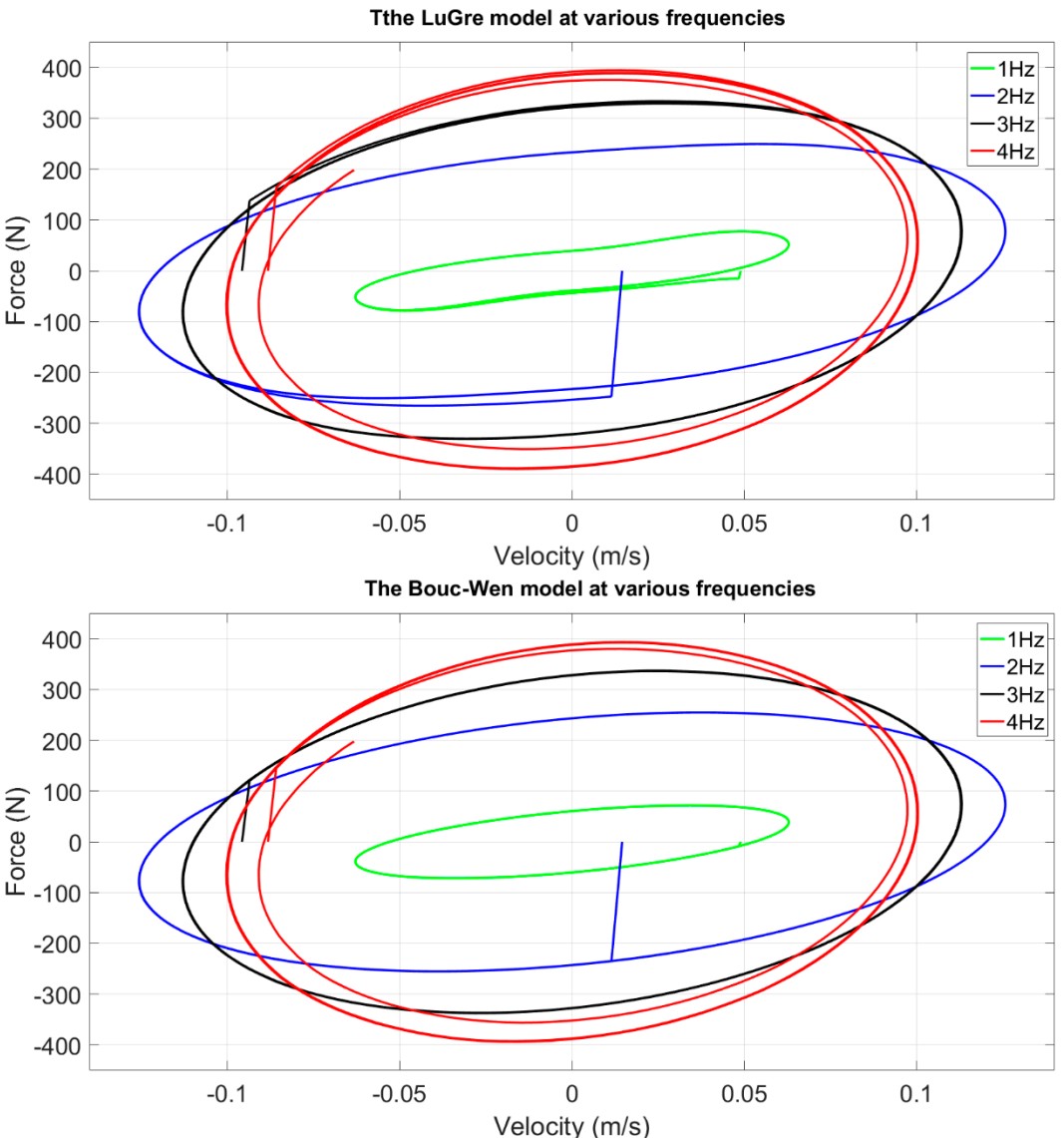

**Figure 15.** Forces vs velocity in various frequencies for different models.

## 6. Conclusions

In this paper, three different dynamic models have been developed for the hysteretic behavior of inerters. This study attempted to provide more realistic models for inerters in comparison with ideal models.

In order to provide experimental data, the performance test was done on a manufactured ball-screw type inerter at different frequencies. The results of the tests were denoised by using EMD. Three different models—theDahl model, LuGre model and Bouc—Wen model—have been introduced to the ideal model of the inerter. The parameters of the models were identified based on the experimental tests. The results of tuned models were compared at different frequencies according to the performed tests. The results demonstrate that the Dahl model cannot predict the behavior of the device in a low frequency range and in the range that the effect of friction is more dominant in comparison of the effect of inertance of the device. However, while the LuGre model has better performance at various frequencies in comparison with the Dahl model, the Bouc–Wen model has the best performance at all frequencies.

The results of this paper provide a good insight into the hysteretic behavior of the ball-screw type of inerter that has not been included in previous research about inerters. These models are far from

ideal models of inerters, and using these models can provide more details about the dynamic behavior of this device in different applications.

**Author Contributions:** H.R.K. has proposed the idea of this work and also the mathematical models. A.S. has done the experiments and developed the analytical models. The remaining part of the work has been done together. All authors have read and agreed to the published version of the manuscript.

**Funding:** This research received no external funding.

**Conflicts of Interest:** The authors declare no conflicts of interest.

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
