# Peer review of "Modelling and Identification of the Hysteretic Dynamics of Inerters"

_designs_

Round 1
Reviewer 1 Report
The paper ‘Modelling and identification of the hysteretic dynamics of inerters’ deals with experimental study on the hysterical behavior of inerters. The paper topic is interesting and the manuscript is well written.
The novelty and original contributions of the proposed paper is very clearly defined. Unlike existing methods that can only consider a constant inertance for static models of the device, we develop three different dynamic models for a ball-screw type inerter. To eliminate the effects of the measurement noise, an empirical mode decomposition method is proposed.
The paper is comprehensive and the results seem coherent and described with sufficient clarity.
The paper can be published after a minor revision:
- References to figures in the text must be replaced (see observation 'Error! Reference source not found').
- Figures 9 - 12 required details for a better view of the three models and the test.
- The text and English language in the revised paper have to checked, edited and corrected by authors preferably by a native English speaker.
Author Response
The authors would like to thank the anonymous reviewer very much for the valuable comments, which lead to significant improvement of the quality and presentation of the manuscript. We have seriously considered these comments and suggestions, and carefully revised the manuscript accordingly. The questions and comments are addressed as follows:
1.The problem associated with the caption of figures was solved. In the revised version, the whole text was checked to be sure that all figures have been referenced appropriately.
2. Based on your valuable comments about Figure (9) to (12), these figures have been plotted again with different formats for the models. In addition, the time histories have been presented for a smaller number of time period. Therefore, in the recent figures it is easier to see and to distinguish various models and their behaviors. .
3. The manuscript was checked again to remove some grammatical mistakes.
In summary, we sincerely appreciate the reviewer for the helpful and constructive comments, which have greatly helped improve the whole level of our paper. We hope the revised version now meets reviewer’s expectation, and thus could be accepted for publication. We would gladly address any other concerns if the reviewer deems it necessary in the next round.

Reviewer 2 Report
- The authors are suggested to proofread the manuscript and rectify the typos and other grammatical mistakes. Also, rectify "Error! Reference source not found" throughout the manuscript.
- What are the conditions deciding the properties of the manufactured inerter? An explanation is required.
- How the associated nonlinearities are compensated? An analysis is required.
Author Response
The authors would like to thank the anonymous reviewer very much for the valuable comments, which lead to significant improvement of the quality and presentation of the manuscript. We have seriously considered these comments and suggestions, and carefully revised the manuscript accordingly. The questions and comments are addressed as follows:
1. The manuscript has been read by the authors and some mistakes have been corrected. In addition, the problem related to the caption of figures has been solved. Therefore, the repeated message in the text; ‘Error! Reference source not found’ has been removed.
2. The properties of manufactured inerter has been considered based on the results of analyses that they have been published in two papers from the authors in machines (2017) and also in Mechanical System and Signal Processing journal (2018). In addition, the details of design and testing process of this device have been explained in another journal paper published in 2018 in designs.
Based on the presented results, the inerter has been designed to change the property of the isolations system in a way that it can protect the slender structure on the isolation system against earthquakes. This idea has been illustrated by performing experiment on a scaled structure (designs, 2018). Also the designed inerter can be used in combination of TMDs in order to increase the performance of the isolation system. This concept has been proven in the published papers based on the results of the model that it was tuned by using experimental tests.
3. The designed and manufactured inerter has been used to change the response of the isolation system in the presence of relatively high pitching motion on the isolation system. Therefore, the nonlinear behavior of inerter (including the effect of backlash) was introduced intentionally to the design phase. In the recent work, we focus on the behavior of the designed inerter in order to have more accurate model for the next application.
It should be mentioned that in the previous papers, the device was modeled as an ideal device. In other words, we just considered the inertance of the designed device in our model, previously.
In summary, we sincerely appreciate the reviewer for the helpful and constructive comments, which have greatly helped improve the whole level of our paper. We hope the revised version now meets reviewer’s expectation, and thus could be accepted for publication. We would gladly address any other concerns if the reviewer deems it necessary in the next round.
